# Revealing Alteration in the Hepatic Glucose Metabolism of Genetically Improved Carp, Jayanti Rohu *Labeo rohita* Fed a High Carbohydrate Diet Using Transcriptome Sequencing

**DOI:** 10.3390/ijms21218180

**Published:** 2020-10-31

**Authors:** Kiran D. Rasal, Mir Asif Iquebal, Sangita Dixit, Manohar Vasam, Mustafa Raza, Lakshman Sahoo, Sarika Jaiswal, Samiran Nandi, Kanta Das Mahapatra, Avinash Rasal, Uday Kumar Udit, Prem Kumar Meher, Khuntia Murmu, UB Angadi, Anil Rai, Dinesh Kumar, Jitendra Kumar Sundaray

**Affiliations:** 1Fish Genetics and Biotechnology Division, ICAR-Central Institute of Freshwater Aquaculture, Bhubaneswar 751 002, India; kirancife@gmail.com (K.D.R.); sangitadixit2011@gmail.com (S.D.); vasam.manohar@gmail.com (M.V.); lakshmansahoo@gmail.com (L.S.); eurekhain@yahoo.co.in (S.N.); kdmahapatra@gmail.com (K.D.M.); avinashrasal44@gmail.com (A.R.); uday.Udit@icar.gov.in (U.K.U.); premmeher@gmail.com (P.K.M.); murmucife@gmail.com (K.M.); 2Centre for Agricultural Bioinformatics (CABin), ICAR-Indian Agricultural Statistics Research Institute, Library Avenue, PUSA, New Delhi 110012, India; ma.Iquebal@icar.gov.in (M.A.I.); mustafaraza@gmail.com (M.R.); sarika@icar.gov.in (S.J.); Ub.Angadi@icar.gov.in (U.A.); anil.rai@icar.gov.in (A.R.); dinesh.kumar@icar.gov.in (D.K.)

**Keywords:** *Labeo rohita*, transcriptome, gene ontology, KEGG, insulin signaling pathway

## Abstract

Although feed cost is the greatest concern in aquaculture, the inclusion of carbohydrates in the fish diet, and their assimilation, are still not well understood in aquaculture species. We identified molecular events that occur due to the inclusion of high carbohydrate levels in the diets of genetically improved ‘Jayanti rohu’ *Labeo rohita*. To reveal transcriptional changes in the liver of rohu, a feeding experiment was conducted with three doses of gelatinized starch (20% (control), 40%, and 60%). Transcriptome sequencing revealed totals of 15,232 (4464 up- and 4343 down-regulated) and 15,360 (4478 up- and 4171 down-regulated) differentially expressed genes. Up-regulated transcripts associated with glucose metabolisms, such as *hexokinase, PHK, glycogen synthase* and *PGK*, were found in fish fed diets with high starch levels. Interestingly, a de novo lipogenesis mechanism was found to be enriched in the livers of treated fish due to up-regulated transcripts such as *FAS*, *ACCα*, and *PPARγ*. The insulin signaling pathways with enriched PPAR and mTOR were identified by Kyoto Encyclopedia of Genes and Genome (KEGG) as a result of high carbohydrates. This work revealed for the first time the atypical regulation transcripts associated with glucose metabolism and lipogenesis in the livers of Jayanti rohu due to the inclusion of high carbohydrate levels in the diet. This study also encourages the exploration of early nutritional programming for enhancing glucose efficiency in carp species, for sustainable and cost-effective aquaculture production.

## 1. Introduction

Aquaculture production is increasing significantly at the global level, contributing to 6.7% of total fishery production as compared to marine production (captured) [1]. For sustainable aquaculture, the availability of the feed and its cost is the greatest concern. During feed formulation, carbohydrate sources are inexpensive and broadly available, and they can be well utilized in aquaculture [1,2]. Earlier studies showed that carbohydrates have great energy value and that the inclusion of carbohydrate sources in fish feed has positive effects in terms of growth and digestibility [3,4,5,6]. However, the metabolism of those feed ingredients is dependent on their source materials, percentage of inclusion, type of fish, and their feeding habits (carnivorous/omnivorous/herbivorous), etc. Thus, the inclusion of a higher level of digestible starch in the diet is economically beneficial in aquaculture [6,7].

Carbohydrates are the main energy source in humans, animals and several vertebrates. In order to determine nutritional requirements, the digestive physiology of commercially important aquaculture species has been reported [7]. Studies have indicated that carbohydrate sources could be incorporated into the fish diet, however glucose homeostasis varies among the teleost species, which leads to changes in the blood glucose level in different fish species [8,9]. Thus, carbohydrate metabolism, which affects glycolysis, glucogenesis, lipogenesis and glycogen metabolism directly or indirectly, was reported by several groups, based on different feed types, and different sizes and species of fish [6,8]. The liver is the largest vital organ, and is associated with important physiological processes such as glycogen synthesis (metabolism), glycogen storage, de novo lipogenesis, detoxification, antioxidant activity, etc. [9]. Gluconeogenesis is an important strategy to regulate the blood glucose level in fish livers, and it may happen due to an inability to utilize carbohydrates [10]. Glucose tolerance tests (GTTs) are commonly used to examine the ability of the fish to use carbohydrates upon inclusion in the fish diets [8]. Regulation of glucose metabolism in the liver by the interaction of the glucose transporter (GLUT) with AMPK and insulin signaling has been depicted by several studies [11]. Digestibility and enzyme activity in response to dietary carbohydrates differ between fish species, and usually depend on the level of dietary intake, and the source and composition of the diet [12,13]. Overall, studies suggest that the livers of most fish species are apparently capable of regulating glucose metabolism and storage. Several hypotheses were proposed for the deficiency of carbohydrate utilization, such as a lack of insulin deficiency, a lack of dependent glucose transporter, and a lack of hepatic glucokinase, but these have been proven wrong [14,15]. In addition to these, it has been reported that temperature and oxygen level are also important factors that affect the metabolic activities of fish [16,17]. However, the mechanism by which insulin regulates plasma glucose levels in the fish liver remains unknown, and the relative contribution of the main sensitive peripheral tissues to this hormone remains still to be clarified. Over the past decade, molecular tools has been used to address some of the downstream components of carbohydrate/glucose metabolism and regulatory processes. Currently, the physiological and molecular basis of this apparent glucose intolerance in aquaculture fish is not fully understood.

The recent progress in sequencing technologies and advancement in computational tools has facilitated an in-depth understanding of gene function/regulation on a genome-wide scale. Large numbers of transcripts, including novel transcripts, have been successfully identified in various organisms using next generation sequencing (NGS) technology, including model as well as non-model aquaculture species [18,19]. In the present study, we have analyzed the transcriptional changes in the livers of farmed rohu, *L. rohita* (Jayanti), upon the inclusion of different levels of carbohydrates in the diet. Farmed rohu is an economically important carp species in South Asia, and it contributes significantly to aquaculture production [20]. Jayanti is an improved variety of *Labeo rohita* produced by selective breeding [20]. The feed cost is the major concern in rohu grow-out culture and feeds with 30–35% protein are required, which in turn increases the production cost [21]. Earlier studies have shown that rohu showed an optimal starch requirement in the diet, of 20–24% [22,23]. Rohu is omnivorous in nature, and supposed to be better at carbohydrate utilization [24]. However, molecular-level investigation with respect to carbohydrate metabolism is lacking in the improved rohu (Jayanti). In this work, we have investigated the molecular mechanism of carbohydrate metabolism in the liver tissues of rohu using NGS technology and computational tools. The generated information will help us to undertake nutritional programming research in rohu as well as other carp species.

## 2. Results

### 2.1. Illumina Sequencing and read Mapping

Through IlluminaHiSeq (2 × 150 bp) pair-end sequence technology, totals of 23,396,281, 24,004,149 and 35,558,985 raw reads were generated in the liver tissue of rohu fish of the 20% (control), 40% and 60% CHO (Carbohydrate) level groups, respectively. After pre-processing and the removal of the adaptor sequences, low-quality sequences and reads with poly-N sequences, totals of 21,202,572, 21,608,231 and 30,800,241 clean reads were obtained in the liver of with the fish in the 20% (control), 40% and 60% CHO level groups, respectively. The obtained raw reads were uploaded to the NCBI SRA database with the following accession numbers: BioSample accession SAMN11246841, SAMN11246839 and SAMN11246840. A total of 88.35% of the reads were successfully mapped to reference genome of rohu. HTSeq v0.6.1 software was employed to measure the number of clean reads mapped to each gene and estimate or calculate transcript abundance. The summary of the analysis of raw and pre-processed read sequences is shown in Table 1.

### 2.2. Analysis of DEGs and Annotation

In order to identify differentially expressed transcripts/genes (DEGs) among the liver transcriptomes of control and high CHO-fed fish, comparative analyses were performed. The differential expression analysis revealed that totals of 15,232 and 15,360 transcripts were differentially expressed in the liver tissues of rohu fish fed with 40 and 60% CHO as compared to the control group (20%). Among that, 4464 transcripts were up-regulated, while 4343 transcripts were found to be down-regulated and 6425 were neutral in the control versus fish fed with 40% CHO. In the fish fed a high CHO (60%) diet, 4478 transcripts were up-regulated, while 4171 transcripts were down-regulated and 6711 were found to be neutral compared to the control (Figure 1). Table 2 describes the number of differentially expressed genes in the control and treated groups. The volcano plot (scatter plot) depicts the distribution of up- and down-regulated transcripts, including neutral expression, in both the control and treated groups (Figure 2). Among these DEGs, a total of 30 were identified as significantly differentially expressed transcripts/genes, and were associated with metabolism, among the control and fish fed a high-CHO diet using the criteria of adjusted *P*-value < 0.05 and (Log_2_ (fold change)) > 1. For the validation of DEGs among control and treated groups, six transcripts/genes associated with metabolism were selected for qPCR, and their relative expression patterns were consistent with the resulting liver RNAseq data. The reads were mapped to a reference database for annotation, and the liver transcriptome data of rohu were annotated with the zebrafish public database (Nr protein, KEGG, GO and COG) using BLAST2GO with an E-value cut-off of <10^−5^. In the BLAST search, gene names and the associated protein accession numbers with respect to transcripts were retrieved, and further gene ontology (GO) terms were assigned. The gene ontology (GO) assignments were performed for classifying the functions of expressed genes/transcripts of the livers of rohu fish based on homology searching. The functional enrichment analysis indicated that all the differentially expressed transcripts of control and treated groups were significantly enriched in three GO terms (Appendix A). In the liver, the 446 up-regulated DEGs found in fish fed a high-CHO (40%) diet were significantly enriched GO terms, with 1749 in the biological processes, 1275 in the cellular components and 2286 in the molecular function categories, while the enriched GO terms in fish fed a high-CHO (60%) diet included 3119 in the biological category, 1278 in the cellular category and 2298 in the molecular function category.

The top gene ontology terms were found to be associated with biological process categories, such as behavior, cell adhesion molecules, threonine phosphatase, developmental process, growth, immunity, localization, locomotion, metabolic process receptor binding, response to the stimulus, insulin signaling, signaling liver development, etc. In the cellular component categories, GO terms were well represented in the cytoplasm, mitochondrion, exocyst, nucleoplasm, organelles, membrane, extracellular region, extracellular matrix, cell, cell junction, synapse, extracellular space, an integral component of the membrane, etc. In the molecular function group, GO terms such as binding, catalytic activity, molecular function regulator, RNA polymerase, motor activity, nucleic acid binding, signal transducer activity, protein binding, protein kinase, transcription cofactor activity, structural constituent of the ribosome, actin-binding and hormone activity were abundantly represented (Figure 3). The Circos3 plot describes overlaps between the differentially expressed genes based on their functions or shared pathways (Figure 4).

### 2.3. Pathway Analysis

KEGG (Kyoto Encyclopedia of Genes and Genomes) pathway analysis is mainly utilized for the systematic representation of gene/transcript functions through their network, which leads to the discovery of gene/transcript functions and their associated metabolic pathways [25]. In this study, the annotated sequences were mapped to the zebrafish reference pathways available in the KEGG database. The sequences were successfully assigned to 433 KEGG pathways in different categories, as shown in Figure 5 and Appendix A. We found that metabolism-associated pathways, such as protein kinases (KID: 01001), the MAPK signaling pathway (04010), G protein-coupled receptors (04030), the insulin signaling pathway (04910), the mTOR signaling pathway (04150), G protein-coupled receptors (04030), the FoxO signaling pathway (04068), lipid biosynthesis proteins (01004), glycolysis/gluconeogenesis (00010) and the PPAR signaling pathway (03320), were the most abundant in control vs. treated groups. As per our experiment on carbohydrate metabolism, we found that 30 genes/transcripts were involved in the glycolysis/gluconeogenesis pathway in the livers of the control vs. 40% CHO and control vs. 60% CHO groups. The *hexokinase, ATP dependent PFK, glycogen synthase, PHK, PGK, ACCα, PPARγ* and *FAS* genes/transcripts were mapped against the insulin signaling pathway (04910), which depicted the involvement of genes in glucose metabolism as well as glycogen synthesis and lipogenesis (Figure 6). The insulin signaling pathways of the liver tissues of the control and treated (high-CHO) rohu fish are depicted in Figure 7.

Clusters of orthologous groups (COG) analyses were carried out to assign or classify orthologous gene products. In this study, totals of 420,648, 409,738 and 489,475 genes/transcripts in the fish of the control 20%, 40% and 60% CHO level groups were clustered into 153, 154 and 164 COG categories, respectively. Among these COGs, most were sequence mapped or assigned to cytoskeleton, general function prediction, translation, ribosomal structure, biogenesis, signal transduction mechanisms, intracellular trafficking, secretion, vesicular transport, etc. The top 26 COG categories in the control and treated groups of the fish are shown in Figure 8.

In the gene regulatory network (GRN), carbohydrate metabolism is coordinated with key candidate genes and hub genes, which were depicted in the GRN (Figure 9). These hub genes included the synaptopodin gene, the LIM and senescent cell antigen-containing domain 1-like protein, reticulon, apolipoprotein, cytochrome P450, parvalbumin alpha, protein O-mannosyl-transferase 2, supervillin-like isoform X2, transient receptor potential, melastatin 4a, myosin, light polypeptide 3, skeletal muscle, guanylate-binding protein 2, phenylalanine hydroxylase complement C4-2, TC1-like transposase, coagulation factor IXb (Zgc:136807), complement factor B/C2A, phospholipid-transporting ATPase, alpha-2-macroglobulin (fragment), S-adenosyl methionine synthase, thrombospondin 2b, notch less homolog 1 (Drosophila), very-long-chain (3R)-3-hydroxyacyl-CoAdehydratase, obscurin-like 1b, and Sk-tropomodulin (tropomodulin 4 (muscle)). The detailed roles of each of the hub genes involved in the gene regulatory network are described in Table 3. Other major functional classes included the improvement of glucose homeostasis, the control of the oxidative glycolytic pathway in carbohydrate metabolism, association with fatty acid uptake by adipocytes, signaling the central hub gene, controlling glucose, and lipid metabolism by insulin. The hub genes in a given GRN have a strong tendency to exhibit pleiotropic effects. A study of these can help identify tissue- and time-specific hub gene regulators.

## 3. Discussion

Recent progress in sequencing technologies has led to improved genomic and transcriptomic resources in various model and non-model species, which has opened doors to address key biological questions. Through omics technologies, several studies reported a molecular-level underpinning of digestive physiology via the inclusion of nutrients and/or other dietary ingredients in the target organs of mammals and aquaculture species. In aquaculture fish, for enhancing production and sustainability, understanding fish nutrition and dietary requirements is a major concern [50]. Carbohydrates are the cheapest source of energy, which determines the input cost of aquaculture production. A few studies revealed that the inclusion of carbohydrates in the diets of fish has positive impacts on aquaculture. Prolonged postprandial hyperglycemia is mainly observed in several fish species (specifically in carnivorous fish) upon feeding with a high-carbohydrate diet [10,51]. The molecular level of glucose intolerance is not understood for several commercially important aquaculture species. Therefore, a study of the system biology network integrated with high-throughput sequencing technologies needs to be applied to delineate the molecular events of key metabolic genes and pathways, which could be used as indicators of carbohydrate/glucose digestibility. In this study, to delineate the molecular level responses of Jayanti rohu, *L. rohita,* upon feeding with different levels of gelatinized starch (20%, 40% and 60%), we performed a total liver transcriptome sequencing to identify the modulation of the key metabolic transcripts/genes and pathways involved in carbohydrate metabolism. Earlier studies hypothesized that the lower metabolic use of enriched carbohydrates/glucose in fish has been attributed to a poor molecular-level regulation of key metabolic genes, as well as their associated pathways, i.e., inefficient glycolysis, inability to produce glucose via gluconeogenesis, and low efficiency of lipogenesis [14,52,53,54].

In this work, the differential expression profiles of all the key metabolic genes/transcripts, changes in the hepatic glucose transporters, and associated pathways, were systematically analyzed using the liver transcriptome data, and they were found to be involved in processes of carbohydrate metabolism such as glycolysis, glycogenesis, gluconeogenesis and lipogenesis (Figure 10 and Appendix A). The liver is a vital organ and the main source of energy for different activities of metabolism, and it synthesizes key enzymes/hormones and transporters [55]. It plays an essential role in glucose homeostasis via the modification/induction of glucose transporters (GT), as well as key enzymes, including the pathways associated with net glucose flux in the hepatocytes, for achieving successful glucose production from the substrate energy yield [10,56].

### 3.1. Alteration in Alpha Amylase and Glucose Transporter Activity

Most fish possess α-amylase (EC 3.2.1.1) activities, which mainly break down the polymers of glucose molecules (α-glycosidic linkages), such as starch/glycogen, but are unable to cleave the β-glycosidicanomeric bonds of some polysaccharides, for example cellulose or hemi-cellulose, etc., which determine CHO digestion efficiency [32]. In the present study, we used a gelatinized starch in the feed preparation, and upon feeding we observed an increase in α-amylase expression activities in the fish fed at a 40% CHO level as compared to the control. This was also increased in the fish fed a 60% CHO diet. Several studies emphasized that the inclusion of a gelatinized starch polysaccharide is the best method to achieve carbohydrate digestibility [32,33]. However, carnivorous fish, such as catfish, exhibited lower amylase activity as compared to herbivorous and omnivorous fish (for example, carp, tilapia) [37].

Several isoforms of the glucose transporters (GLUT) are reported to act as glucosensing systems, and play a vital role in the transport of glucose molecules into hepatocytes or muscle cells during carbohydrate metabolism mediated by insulin [53,57,58,59]. There are 14 members of the glucose transporters/GLUT protein family [60,61]. In our studies, we observed that hepatic glucose transporter 2 and solute carrier family 2 (facilitated glucose transporter), member 8 *(slc2a8)*, activities were increased in the livers of fish fed a high-CHO diet as compared to the control, while the sodium-dependent glucose transporter 1 expression was neutral in both groups. However, recent studies have shown that modulations of the gene expression activities of glucose transporter 1 occur in grass carp, *Ctenopharyngodon idellus*, fed with high dietary carbohydrate levels [62]. Contrastingly, hepatic *GLUT1* and *GLUT2* were dysregulated in tilapia after 1–3 h due to the inclusion of high CHO in the diet, while an increase in the plasma glucose level was observed [63]. In carnivorous fish, such as rainbow trout and Atlantic cod, the expression level of hepatic glucose transporters 2 (*GLUT2*) did not rely on the feeding regime (i.e., feeding or fasting status), but hepatic *GLUT2* and muscle-specific *GLUT4* gene expression were stimulated in the rainbow trout during re-feeding [64,65,66,67].

Recent studies have implied that the down-regulation of hepatic *GLUT2* gene activity occurred due to the inclusion of high CHO in the diet of Japanese flounder and rainbow trout, and thereby reduced the activity of the PI3K-AKT signal transduction pathway and glycolysis [68,69]. The present study is also in line with an earlier report, which demonstrated that high dietary CHO assimilation was inhibited due to the failure of mechanistic glucose transportation across hepatic cells [6]. In higher vertebrates, the hepatic *GLUT4* gene expression in hybrid giant grouper was increased due to the inclusion of digestible CHO, such as maize starch, in the diet [70]. In our study, we did not observe any expression transcripts of *GLUT4* in any of the treated groups compared to the control.

### 3.2. Modulation of the Glycolytic Mechanism

The metabolic activities in the liver are tightly controlled and mediated by the insulin signaling pathway, which in turn maintains glucose homeostasis [55,71]. In the liver, glucose is taken up by hepatocytes with the help of glucose transporters, and further converted into pyruvate via glycolysis. ATP energy generation, followed by the TCA cycle and further excesses of CHO, induces the process of glycogen synthesis and de novo lipogenesis. The key regulatory enzymes, such as *glucokinase*, (*GK/GCK*) or *hexokinase* (*HK*), *pyruvate kinase* (*PK*) and *phosphofructokinase* (*PFK*), play a vital role in glycolysis in the liver [72,73,74]. The inclusion of high levels of CHO/glucose in the diet enhanced *GK* gene expression in several aquaculture species, such as rainbow trout [64], which is the first key enzyme in the glycolysis process.

In our study, we also observed a heightened expression of the *hexokinase 1* gene in the 40% CHO-fed fish as compared to the control; contrastingly, *HK1* expression was down-regulated in the 60% CHO-fed fish as compared to the control. Similarly, the *phosphofructokinase* (*PFK*) expression level showed similar trends to *hexokinase* in all treatment groups. *GK* and *PFK* are two key indicators for effective hepatic glycolysis in several fish, such as tilapia [53,63]. Recent studies showed increases in the *GK, PK, PFK* and *G6PDH* gene activities in the livers of *Megalobrama amblycephala* due to the incorporation of high CHO levels in the diet [62,75,76], and similar trends were observed in other species, such as gilthead sea bream (*Sparus aurata*) [71], rainbow trout (*Oncorhynchus mykiss*) [10,12,77,78], tilapia [63], giant grouper (*Epinephelus lanceolatus*) [70], juvenile largemouth bass, and *Micropterus salmoides.* Subsequently, we observed an increased level of expression of the key enzymes of glycolysis in the control vs. fish fed a 40% CHO diet, while they were down-regulated in the control vs. fish fed a 60% CHO diet. The enzymes associated with gluconeogenesis were found to be down-regulated in both the groups of fish.

The *pyruvate kinase (PK)* gene is the major key regulatory enzyme of the glycolytic pathway, and studies have reported an increase in its expression profile due to high CHO levels in the diets of fish [79] such as perch (*Perca fluviatilis*) [80], gilthead sea bream (*Sparus aurata*) [81], European sea bass [82], gibel carp (*Carassius auratusgibelio*) and Chinese long snout catfish (*Leiocassis longirostris gunther*) [83]. The hepatic *PK* gene expression activities were enhanced significantly in grass carp with an increase in the percentage of CHO levels in the diet [74]. Interestingly, our results showed a neutral expression profile for the *PK* gene in all the treatments, including the control. In common carp (*Cyprinus carpio*), *pck* transcription was strongly inhibited as a result of feeding [84], while *g6pc* expression was sharply down-regulated after feeding in gilthead sea bream [64,84].

### 3.3. Glycogen Synthesis Enrichment due to High CHO in the Diet

It has been reported that excess glucose available from the diet is stored in the form of glycogen in the liver through the process of glyconeogenesis [56,85,86]. We observed an increase in the *glycogen synthase (GYS)* and *UDP-Pyrophosporylase* expression levels, possibly due to induction in the process of glycogenesis in the liver of rohu for glycogen storage. This trend was similarly reported in *Megalobrama amblycephala*, gilthead sea bream and golden pompano (*Trachinotus ovatus*) [75,77,87] when fed a diet with increased CHO levels. In tilapia, glycogen accumulation in the liver is induced with the help of the key gene *glycogen synthase (GYS)* [63].

### 3.4. Impact of High-CHO Diet on Gluconeogenesis and Lipogenesis

Gluconeogenesis is the reverse process of glycolysis, which produces glucose using pyruvate as a substrate. Several studies reported that this process is only activated during fasting challenges in fish for ATP energy production. Excess glucose availability in the liver represses the enzymatic activities of gluconeogenesis. The regulatory enzymes include *fructose bisphosphatase (FBPase)* and *phosphoenolpyruvate carboxylase (PEPCK-C),* which mainly control the key steps of gluconeogenesis. A higher level of *FBPase* gene expression was observed in fish fed with a starch diet, namely, European sea bream and gilthead sea bass [10,82]. A down-regulation of the key enzyme *G6Pase*’s activity in the gluconeogenesis pathway has been observed in *Megalobrama amblycephala* fed with dextrin [88], and this was up-regulated during fasting in the livers of rainbow trout [89,90,91]. The hepatic gluconeogenesis pathway (*PEPCK* mRNA expression and activity, and *G6Pase* activity) in zebrafish, *D. rerio* adults, was repressed by feeding them with early high CHO after CHO intake [92]. In the juvenile golden pompano (*Trachinotus ovatus*), the endogenous glycolysis pathways were enhanced, while the gluconeogenesis process was down-regulated by an increase in the dietary CHO level [93].

In the liver, excess amounts of carbohydrates (pyruvate) convert into fatty acid via the de novo lipogenesis pathway [7]. The acetyl-CoA produced through the citrate acid cycle converts into malonyl-CoA using acetyl-CoA carboxylase (ACC), and then palmitic acid is produced by fatty acid synthase (FAS) with NADPH as a precursor. It has been reported that de novo lipid synthesis is modulated by enhancement of the dietary carbohydrate level or increases in the glucose load in the diets of fish [75,94,95,96,97,98,99,100]. Here, *fatty acid synthase (FAS)* plays a critical role in the process of de novo lipid synthesis. This study also showed an increase in the expression levels of *fatty acid synthase (FAS), acetyl-CoA carboxylase α* and *malonyl-CoA* in fish with higher CHO levels in the diet compared to the control group. In blunt snout bream, *Megalobrama amblycephala*, the hepatic *FAS, acetyl-CoA carboxylase α, delta-6 fatty acyl desaturase,* and *peroxisome proliferator-activated receptor γ* (*PPARγ*) showed elevated expression levels, which indicates induction of the lipogenesis pathways in the liver due to increases in CHO level in the diet [75]. Recently, a study on the liver of tilapia showed the up-regulation of transcripts such as *accα, fas* and *dgat2* due to the incorporation of high CHO levels in the diet, but hepatic *lpl* and *fatp5* expression showed no significant consequences [98,101,102,103].

The heightened level of *cd36* gene expression was also observed in large yellow croakers fed with high-CHO (starch) and low-lipid diets, which implied that *cd36* is crucial for lipid homeostasis via regulation of the PPAR pathway [104]. Elevated levels of hepatic *accα, fas* and *dgat2* gene expression have been demonstrated in rainbow trout (*Oncorhynchus mykiss*) due to high levels of dietary CHO carbohydrates [105,106]. The induction of lipogenesis was observed in our study, with evidence of associated elevated gene expression, which may not be directly linked with glucose metabolism, but the enriched pathways implicated their role in glucose homeostasis via converting excess glucose into fatty acids.

### 3.5. Immune-Associated and Stress-Related Transcriptional Changes in the Liver due to the Inclusion of a High-CHO Diet

It was plausible to enhance the stress response and immunity-related gene modulation in fish fed with a formulated diet [107]. It has been reported that the cortisol level increased in *M. amblycephala* and *Erythroculterilishaeformis bleeker* [3] due to the high CHO level in the diet, which indicated stressful conditions for the fish. Similarly, plasma lysozyme activity decreased in several fish, with an increase in dietary CHO, of the species *Epinephelus malabaricus* and *M. Amblycephala* [108]. The down-regulation of constitutive immune genes was observed in the liver of Atlantic salmon under starvation using a transcriptome study [107]. At the molecular level, heat shock proteins (HSPs) are stress indicators that are involved in stress response, cell protection and improved tolerance level [109]. In rainbow trout, higher levels of CHO increased the *HSP70* gene expression [73] and levels of hepatic *HSP70* in Wuchang bream (*M. amblycephala*) fed with 53% CHO compared to control [110]. In contrast to this, we could not observe any differential expression pattern of *HSP70* among control and treated rohu fish. The *HSP70* transcript was detected only in fish fed with the 40% and 60% CHO-level diets. Interestingly, fish fed with high percentages of CHO (40% and 60%) showed an increase in *HSP90* and associated transcription factors, which act as chaperones and folding catalysts, such as *hsp90b1, dnajb1b, hspa4a, hspb7/8*, etc. Specifically, we detected higher expression levels of *Activator of HSP90 ATPase homolog 1, DnaJ (HSP40) homolog* and *HSP60* in the fish fed a 60% CHO diet compared to the control group counterpart, which indicated the emergence of stress in liver cells.

We detected increased expressions of cytokine receptors such as *IL-6* and *suppressor of cytokine signaling 3* (*SOCS3*) in the livers of fish fed with high CHO as compared to the control. This is in line with earlier studies, in which the augmented levels of the expressions of *IL-6* and *SOCS3* resulted in chronic stress, as well as impacteing the glucose transport system of Japanese flounder (*Paralichthys olivaceus*) fed with high dietary CHO [68]. Recent studies revealed that immune-associated transcriptional changes in the liver of *Epinephelus akaara* occurred due to a high level (30%) of carbohydrates in their diet, which in turn resulted in the over-expression of immune-related genes such as *CXCR4, CCL4, IL8, TLR9* and *NFκB inhibitor alpha,* etc. [111]. In this study, the genes associated with the chemokine signaling pathway, such as *C-C motif chemokine 4 (CCL4), C-X-C chemokine receptor type 4 (CXCR4)* and *C-X-C motif chemokine 10 (CXCL10)*, were significantly expressed in the group fed a high-carbohydrate diet.

A high CHO level can lead to the over-expression of immune-related genes, which may activate acute inflammatory processes in the liver. Toll-like receptors (TLR), tumor necrosis factor (TNF), interferon regulatory factor 3 (IRF3) and tumor necrosis factor receptor (TNFR) belonging to the Toll-like receptor signaling pathway showed differential levels of expression patterns in fish fed with different levels of CHO in their diet. *TLR3, TLR5, IRF3, IRF5* and *IRF5* were up-regulated in the livers of fish fed with high CHO levels compared to the control group. This is in line with earlier reports, which demonstrated that modulation of the liver transcripts was associated with the Toll-like receptor signaling pathway in fish fed a high-carbohydrate diet [111]. Interestingly, some of the TLR isoforms, such as *TLR22, TLR1* and *TRAF3,* were found to be down-regulated in the livers of fish with high CHO levels in their diet. Subsequently, our analysis of fish fed with high levels of CHO reported the up-regulation of transcripts such as *Death-domain-associated protein, Caspase-6, Lamin B3, Cathepsin Z, DAB2-interacting protein b*, and *Membrane protein, palmitoylated 5b (MAGUK p55 subfamily member 5)*, which are associated with the apoptosis pathway, possibly inducing hepatic cell death through the extrinsic pathway of programmed cell death (PCD) via the modulation of the p53 signaling pathway. The incorporation of dietary carbohydrates may lead to stress as well as lower immunity through the innate and adaptive immune system. Overall, the up-regulation of a large number of immunity- and stress-related genes in the immune system pathway revealed that the rohu fish’s liver may experience an inflammatory response. The stress in the hepatic cells occurred due to the excess levels of glucose, via modulation of the isoform of the enzymes. Overall alteration in metabolic pathways may occur in liver tissues due to the effects of temperature or oxygen levels on the metabolism of fish, which is a vital area of research and is lacking in the present work.

### 3.6. Insulin Signaling Pathways, mTOR, and the PPAR Signaling Pathway

The insulin signaling pathways are mainly involved in maintaining glucose homeostasis via the uptake of glucose for substrate energy generation, as well as excess glucose storage in the liver (in the form of glycogen via glycogenesis) and the induction of de novo lipid biosynthesis. After the consumption of carbohydrates, insulin promotes the entry of blood glucose using glucose transporters via modulating various series of genes, including GTs, insulin receptors, IRS and hexokinase, and activates the IRS-PI3k-Akt pathways. In our study, we observed active insulin signaling pathways in the livers of fish fed a high-CHO diet, which is in line with earlier reports (Figure 7). In this pathway, INS actions are mediated via insulin receptors (receptor tyrosine kinase family), and then by further modulating pathways [112]. High CHO levels in the diet result in hyperglycemia in rainbow trout, which activates the components of the insulin-signaling cascade. The modulation of the insulin signaling pathway (Akt/TOR) and a postprandial expression pattern of the genes related to metabolism were observed in rainbow trout fed with different carbohydrate to protein dietary ratios [78].

We observed increased levels of expression of key genes of the insulin signaling cascade, such as *phosphatidylinositol 3-kinase (PI3K)*, which subsequently phosphorylates other proteins such as AKT, also known as protein kinase B. This cascade (IRS-PI3K-AKT) further modulates the mTOR pathway, which activates other cascade molecules regulation, such as *ribosomal S6 kinase protein (S6K1)*, and this leads to an increase in metabolism and cell growth. In our study, heightened expression levels of insulin receptor substrate (IRS) were observed. We highlighted the up- and/or down-regulation of important genes involved in the insulin signaling pathway. Our study implied that the up-regulation of *GLUT2* mediated glucose uptake in the livers of fish. Earlier studies also indicated the modulation of *Akt* for insulin signaling, and *S6k* for mTOR, which regulates the metabolic pathways in the liver [69]. In contrast, in our study, the *Akt* and *S6k* expressions were not significant in fish fed high-CHO diets. Recently, researchers have depicted the up-regulation of *S6K1* and *IRS-1* in the livers of blunt snout bream due to an increase in arginine level (2.70%), which indicated a controlled expression of insulin [113]. In this study, we found a neutral expression of *IRS* and *PI3K* in the fish fed a high-CHO diet, and in control fish.

Lipid homeostasis is maintained through key enzymes under the control of *PPARα* (regulation of fatty acid catabolism) and *PPARγ* (regulation of fatty acid storage), which belong to the nuclear receptor super family. We observed increased levels of expression of PPAR enzymes, which enhances the lipid biosynthesis mechanism. Researchers have suggested that treatment with bovine insulin mediated the up-regulation of *PPARγ* and *PPARα* genes in yellow catfish [114,115]. In tilapia, the modulation of peroxisome proliferator-activated receptor (PPAR) and the insulin signaling pathways was observed for de novo lipogenesis due to high dietary CHO levels [101]. The AKT–mTOR pathway also represents an important pathway during metabolism, as reported in Senegalese sole fed with different dietary protein to carbohydrate ratios [116,117]. It has been reported that the de novo lipogenesis mechanism was enhanced due to the modulation of genes such as sterol regulatory element-binding protein 1c (SREBP1c) and TORC1 [118,119], and that transcripts linked with lipogenesis and SREBP1 were up-regulated via the mTOR-dependent pathway. In this study, the *FAS, ACCα* and *PPARγ* genes associated with de novo lipogenesis were found to be up-regulated significantly in the livers of fish fed a high CHO diet compared to control fish, but we could not detect transcripts for the sterol responsive element binding protein (SREBP1) in the livers of either control or treated fish. This clearly indicates that the inclusion of high CHO levels in the diet can promote de novo fatty acid biosynthesis and fat accumulation in the livers of fish via activation of the mTOR pathway.

## 4. Materials and Methods

### 4.1. Experimental Diet Preparation and Feeding Experiment

The experimental feed was formulated in our laboratory and contained three different levels (20%, 40%, and 60%) of dietary carbohydrates (gelatinized starch) (Table 4). The 20% carbohydrate diet was considered as the control and the 40% and 60% carbohydrate levels were designated as high-carbohydrate diets (CHO). Fish meal and casein served as protein sources and corn starch was used as the main source of carbohydrates. The corn starch was gelatinized with the addition of water in an autoclave at 120 °C for 1 hr. All the ingredients were weighed and mixed properly with the addition of oil and water. The prepared dough was pelleted using a hand pelletizer and dried in hot air over night at 60–80 °C. The dried pellets were broken and sized using a grinder into pieces with a 2–3 mm diameter and stored in polyethylene bags at room temperature (30 ± 2 °C) for further use. The experimental fish (40–50 g) were collected from nursery ponds of the Division of Fish Genetics and Biotechnology, Central Institute of Freshwater Aquaculture, Bhubaneswar, Odisha, India. All the experimental procedures were performed with the approval of the Institute Bio-Safety Committee (IBSC) on Ethics. The procured fish were acclimatized in the circular tanks (1000 L) of the wet-laboratory for a period of one week. After acclimatization, the fish were randomly distributed in experimental tanks (500 L) with 10 fish per tank. The experiment was performed in triplicate and the respective diets were fed twice at 2–3% body weight. The water was partially changed and refilled with fresh water. During the feeding trial, the experimental fish were reared with optimum water quality parameters, with water temperature 27–31 °C, pH 7.3–7.8 and dissolved oxygen 5.0–6.0 mg/L, measured as per standard procedure (APHA 1985). After 45 days of the experiment, liver tissue samples were collected from three experimental fish per each treatment (total 12 fish from each treatment). The fish were anesthetized in a water tank containing 100 mg/L of MS-222 (tricainemethanesulfonate, Sigma, Milwaukee, WA, USA). The liver tissue samples were immediately collected and frozen in liquid nitrogen (LN2) and subsequently stored at −80 ˚C for further processing for RNA isolation.

### 4.2. RNA Isolation and cDNA Library Preparation, NGS Sequencing

Total RNA from liver tissues was isolated using an RNAiso kit (TakaRa, Clontech, Mountain View, CA, USA) according to the manufacturer’s instructions. The three fish livers from each treatment tank were pooled for total RNA extraction. The concentrations of total RNA were checked using a Qubit 2.0 Fluorometer (Life Technologies, Carlsbad, CA, USA). The quality of RNA was assessed using the Agilent2100 Bioanalyzer system (Agilent Technologies, Santa Cruz, CA, USA) and those samples with an RNA Integrity Number (RIN) > 7.0 were utilized for Illumina RNA library preparation. Nine cDNA Illumina libraries were generated from the total RNAs of liver samples. RNA was fragmented using nebulization techniques using the NEB Next First Strand Synthesis Reaction Buffer (5 ×), followed by first strand cDNA synthesis with the help of hexamer primers and M-MuLV Reverse Transcriptase, then subsequently synthesized into a second strand of cDNA. The Illumina adaptors were used for the preparation of the library and purified using Agencourt Ampure XP SPRI beads, then further enriched using PCR amplification for cluster generation. The cDNA libraries were generated using the NEBNext Ultra directional RNA Library Preparation kit (Illumina) as per the manufacturer’s protocol and the quality of the library was assessed using the Agilent 2100 Bioanalyzer system (Agilent Technologies, Santa Cruz, CA, USA). Then, all the treatment and control group libraries were pooled for sequencing. The adaptor-ligated libraries of rohu liver samples processed for sequencing using an Illumina Hiseq 500 platform and paired-end reads (150 bp) were generated. The generated raw reads of rohu liver samples were submitted through the NCBI SRA, and are accessible via the NCBI BioProject accession.

### 4.3. Quality Check, Assembly and Mapping

The generated raw reads were further checked for quality using FastQC software (http://www.bioinformatics.babraham.ac.uk/projects/fastqc). Initially, the raw reads were pre-processed to remove the adapter sequences and the low-quality bases (<q30). The Cutadapt2 algorithm was employed to remove reads with adapters using Python scripts. The quality check and filtered reads were further subjected to HISAT software for alignment, and the clean reads were aligned to the reference genome (PRJNA437789) with default parameter sets. Based on that, reads were classified into aligned reads (aligned to the reference genome) and unaligned reads. The cleaned aligned reads were mapped to multiple reference genes.

### 4.4. Gene Regulatory Network

Cytoscape (version 3.7.2) was used for the analysis of the gene regulatory network between the 20% (control), 40% and 60% (carbohydrate regime) groups. Carbohydrates related to up- and down-regulated genes were selected for network construction. Algorithms like ARACNE (algorithm for the reconstruction of accurate cellular networks) and the Network Analyzer plug-in were used for analyzing the gene regulatory network of all three sets of DEGs. On the basis of high degree and betweenness, hub genes were selected. The genes within a network with a higher connectivity played a central regulatory role, and were important in detecting highly connected genes in a network.

### 4.5. Differential Expression Analysis of Transcripts, Gene Ontology and KEGG Pathway

Here, the transcripts were annotated using a homology approach to assign functional annotation with BLAST2GO. Transcripts were assigned with a homolog protein from other organisms if the match was found with an E-value 1.0 × 10^−5^ and minimum similarity greater than 30%. HTSeq v0.6.1 was utilized to measure the number of clean reads mapped to each gene, and estimate or calculate their abundance. The transcripts were normalized using the standard method—the number of count genes divided by the number of FPKM (fragments per kilo base per million reads) of transcripts for each sample. The absolute counts for each transcript were identified and were used in differential expression calculations. In order to check the differential expressions of those transcripts in fish fed a high-carbohydrate diet (40% and 60% CHO) compared to control groups (20%), DESeq was used to calculate the differentially expressed transcripts, and transcripts were categorized into up-, down- and neutral-regulated based on the log2fold change cut-off of 1 value. The transcripts/genes recognized by DESeq with an adjusted *P*-value < 0.05 were categorized as differentially expressed genes (DEGs). The *P*-values of differentially expressed transcripts were adjusted using the Benjamini–Hochberg method to control the false discovery rate. Here, the difference in the expression levels of transcripts was estimated between the control group and fish fed a high-carbohydrate diet.

The functional annotations were performed by homology searching against the public databases, such as the non-redundant nucleotide (Nt) and non-redundant protein (Nr) database of NCBI (http://www.ncbi.nlm.nih.gov/), Swiss-Prot (http://www.ebi.ac.uk/uniprot/), clusters of orthologous groups (COG, http://www.ncbi.nlm.nih.gov/COG/), Gene Ontology (GO, http://www.geneontology.org/), and the Kyoto Encyclopedia of Genes and Genomes (KEGG, http://www.genome.jp/kegg/). The results obtained through BLAST2GO were further subjected to WEGO software for classification of Gene Ontology (GO) categories such as biological processes, molecular functions, and cellular components (http://wego.genomics.org.cn/cgi-bin/wego/index.pl). The GO and KEGG analyses were carried out to find the differential expressions of transcripts/genes involved in various enriched metabolic pathways. The GO enrichment analysis was performed using the R-package (Bioconductor) software EdgeR (Empirical Analysis of Digital Gene Expression in R, http://www.bioconductor.org/packages/2.12/bioc/html/edgeR.html). The categorized GO terms were ranked as per their corrected *P*-values < 0.05. The metabolism-associated KEGG pathway analysis was carried out using online KEGG Automatic Annotation Server (KAAS) software (http://www.genome.jp/tools/kaas/). The Bi-directional Best Hit (BBH) method was employed in KEGG pathway identification. Due to the absence of an annotated genome of rohu in the database, we used Zebrafish (*Danio rerio*) as a reference organism for pathway identification. The pathways per transcript were mapped to differentially expressed transcripts. The relevant metabolism associated pathways were significantly enriched based on the false discovery rate (FDR) ≤ 0.05 of regulated genes. The COG database (clusters of orthologous groups) was used to detect the putative function of transcripts based on known or conserved orthologous genes or their products.

### 4.6. Quantitative Real-Time PCR (qPCR)

In order to validate the generated liver transcriptome data, we selected six significant transcripts/genes associated with carbohydrate metabolism of rohu for performing quantitative real-time PCR (qPCR) analysis. The primers were designed using FastPCR (version 5.2.113) software for the selected transcripts and one housekeeping gene (β actin). The cDNA of the total RNA pool of the liver tissues of fish was generated using a BluePrint^TM^first Strand cDNA Synthesis kit (TakaRa, Clontech, Mountain View, CA, USA). The primer sequences were found in the total RNA from the liver tissues (in triplicate from each treatment) of the control fish and fish treated with high CHO levels, extracted using the RNAiso kit (TakaRa) according to the manufacturer’s instructions. The RNA samples were taken as three technical replicates for performing qPCR. The quantitative PCR was carried out using a Light Cycler-480 SYBR Green I kit (Roche Diagnostics, Mannhein, Germany) as per the manufacturer’s instructions. The PCR mixture consisted of 20 μL, which included a 2 μL DNA template, 0.4 μL each of forward primer and reverse primer, 10 μL SYBR^®^Premix and 0.4 μL ROX Reference Dye (50×), and the total volume was made up with RNase-free water. The qPCR analyses were performed with the following conditions: initial denaturation at 95 °C for 30 s, then followed by 35–40 cycles of denaturation at 95 °C for 10 s, annealing at 60 °C for 30 s, and extension at 72 °C for 30 s. The difference between the cycle threshold (ct) value of selected genes and that of the internal control housekeeping gene was calculated and then the relative gene expression was measured using the 2^−ΔΔCt^ method. The box plot was used to analyze the variation distribution in the normalized count data. A Pearson’s correlation coefficient analysis was carried out to check the co-linearity of count data among control and treated fish. The significant difference between control and high-CHO-fed fish was analyzed using a Student’s paired t test and a one-way ANOVA test (significance value; *P* ≤ 0.05).

## 5. Conclusions

In order to understand the regulatory mechanisms of carbohydrate metabolism in the liver, Jayanti rohu fish fed a high-carbohydrate/glucose diet were analyzed using a series of key genes/transcripts associated with hepatic glucose transport and assimilation. The liver is a major organ in fish, and it indicates the nutritional and physiological status of fish. Hepatic genes were modulated due to the increased dietary carbohydrate levels in the livers of fish fed with a higher proportion of CHO in the diet. The present study suggested that the inclusion of high levels of CHO in the diet promotes glucose transport activities, glycolysis, glycogenesis, lipogenesis, and the pentose phosphate pathway. The transcripts associated with glycogen synthesis, such as *hexokinase, glycogen synthase, PHK* and *PGK*, were found to be up-regulated in the livers of fish fed with high levels of CHO as compared to the control, indicating the enrichment of the glycogenesis mechanism in the liver. In addition to this, the increased levels of expression of *ACCα, PPARγ* and *FAS* in the livers of fish fed with high levels of CHO indicated the activation of de novo fatty acid biosynthesis. Altogether, we have highlighted the enhanced glycolytic process as facilitated by the uptake of glucose and further activated or mediated by the insulin signaling pathway. We also identified enrichment of the insulin signaling pathways and the peroxisome proliferator-activated receptor (PPAR) pathway for de novo lipid biosynthesis. This study emphasized the fine balance of the ratio of dietary carbohydrates to protein in fish, and that the best available energy sources should be used. This work also encourages the investigation of the impact of temperature or oxygen levels on the metabolic pathways of rohu. Overall, this work suggests the incorporation of a higher level of digestible carbohydrate (gelatinized starch) in the diets of rohu, and encourages the undertaking of nutritional programming research in carp aquaculture.

## Figures and Tables

**Figure 1 ijms-21-08180-f001:**
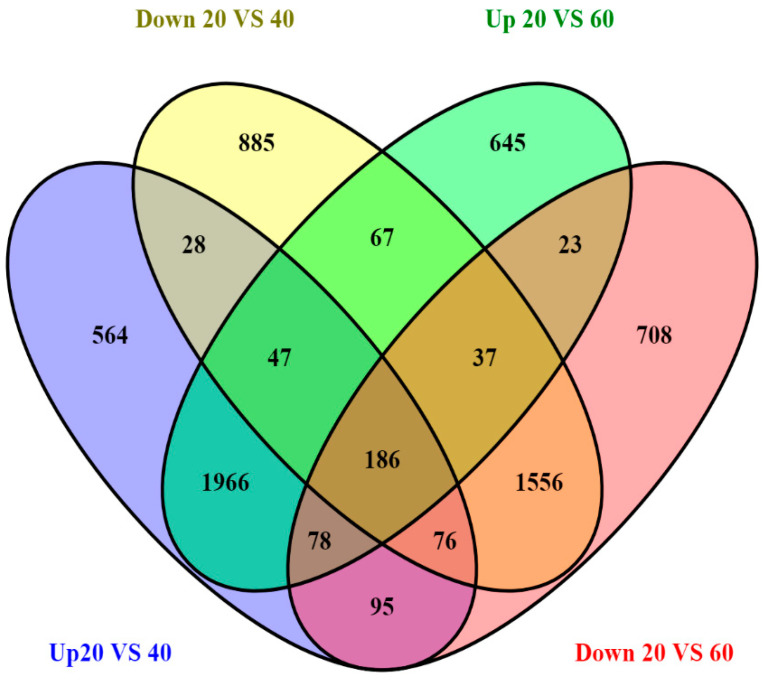
Venn diagram of differentially expressed genes. The sum of numbers in each big circle is the total number of differentially expressed genes in each comparison group, and the overlapping part is the number of common differentially expressed genes among the comparison groups.

**Figure 2 ijms-21-08180-f002:**
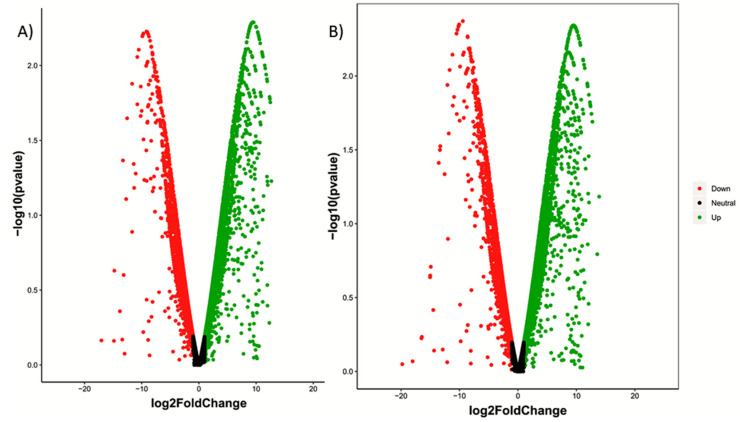
Volcano plot for differential gene expression. Fish feed with (**A**) 40% CHO vs. 20% CHO and (**B**) 60% CHO vs. 20% CHO. Scattered points represent genes: the x-axis is the fold change for the ratio of treated vs. control, whereas the y-axis is the statistic or -Log 10 (*P*-value), which shows the probability that a gene has statistical significance in its differential expression. The green dots are thus genes significantly up-regulated after treatment, red dots are genes significantly down-regulated after treatment and black dots are genes significantly neutrally regulated after treatment.

**Figure 3 ijms-21-08180-f003:**
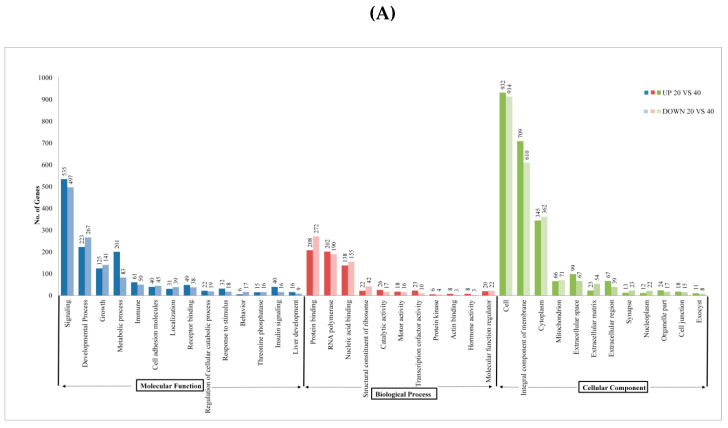
Gene ontology distribution (CHO-related) for the molecular function, biological process and cellular component, of (**A**) Control versus fishes fed with 40% CHO, and (**B**) control versus fishes fed with 60% CHO. The y-axis indicates the number of genes annotated to one GO term. The up-regulated genes under Molecular Function are shown as blue and light blue represents down-regulated genes, biological process up-regulated genes are shown as red and down-regulated genes are shown as light red, and cellular component genes are shown in green for up-regulation and light green for down-regulation.

**Figure 4 ijms-21-08180-f004:**
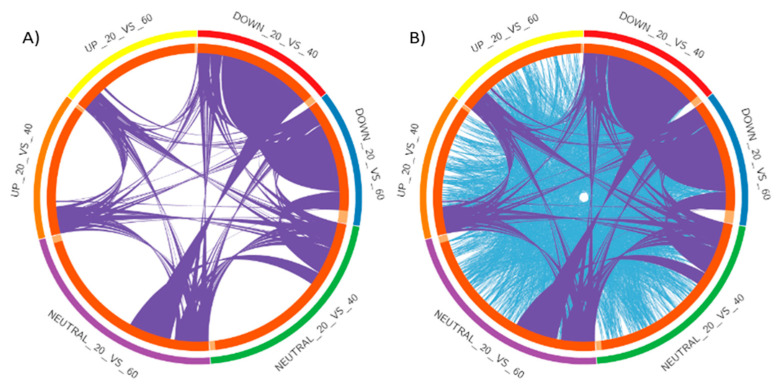
Circos plot showing overlap between gene lists (**A**) only at the gene level, where purple curves link identical genes; (**B**) including the shared term level, where blue curves link genes that belong to the same enriched ontology term. The inner circle represents gene lists, where hits are arranged along the arc. Genes that hit multiple lists are colored in dark orange, and genes unique to a list are shown in light orange.

**Figure 5 ijms-21-08180-f005:**
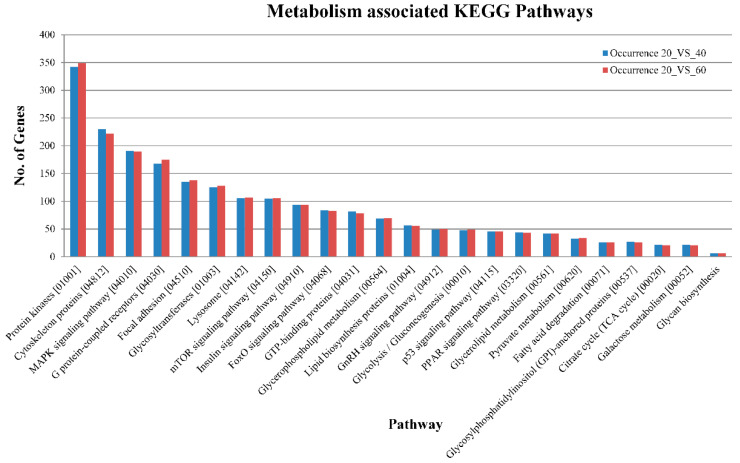
Kyoto Encyclopedia of Genes and Genomes (KEGG) pathway enrichment analysis for selected genes between control (20%) versus fish fed with 40% CHO (blue), and control (20%) versus fish fed with 60% CHO (red), represented on a graph, with pathways on the x-axis and number of genes on the y-axis.

**Figure 6 ijms-21-08180-f006:**
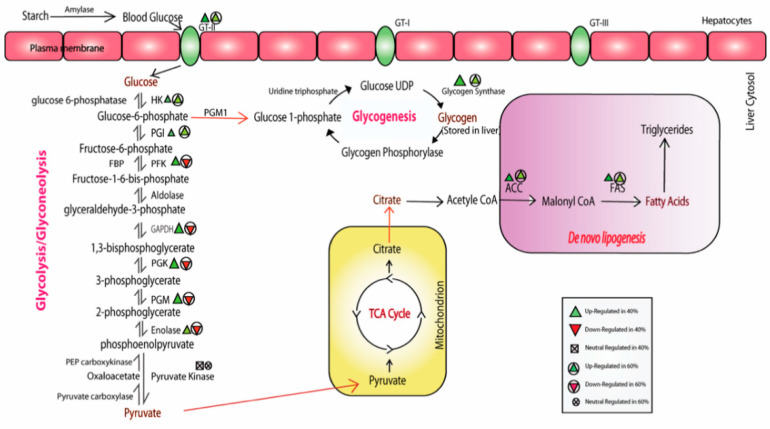
Up-regulation and down-regulation of key metabolic genes in hepatocytes.

**Figure 7 ijms-21-08180-f007:**
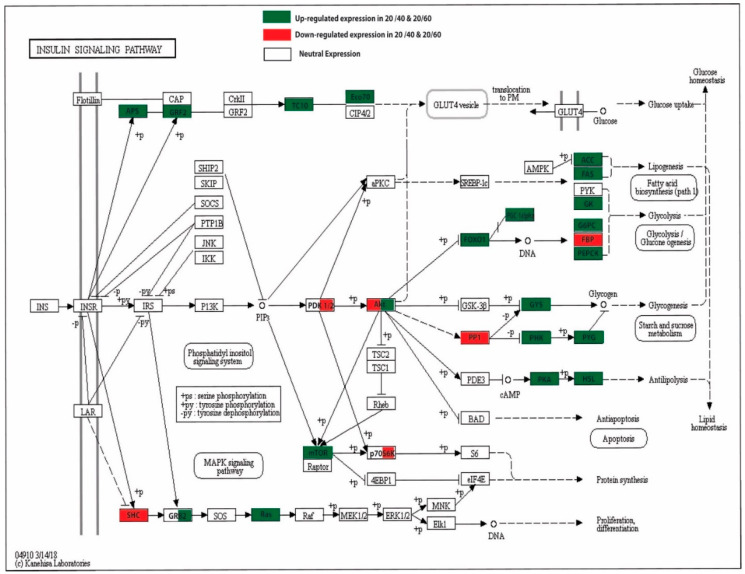
Insulin signaling pathway. Key components of pathways are depicted, red indicates down-regulation, green indicates up-regulation and black indicates neutral regulation of key genes/enzymes.

**Figure 8 ijms-21-08180-f008:**
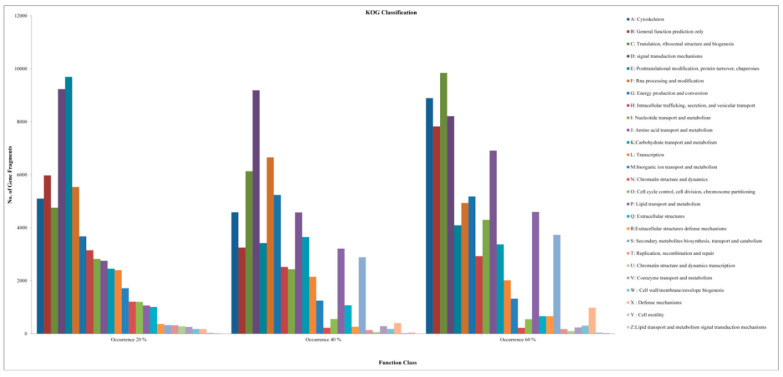
COG functional classifications of *Labeo rohita* fish fed with 20% (Control), 40% and 60% CHO. In total, 26 selected KOG categories were taken for the histogram. The x-axis indicates KOG categories on the right side of the histogram; the y-axis indicates the number of gene fragments in each functional cluster.

**Figure 9 ijms-21-08180-f009:**
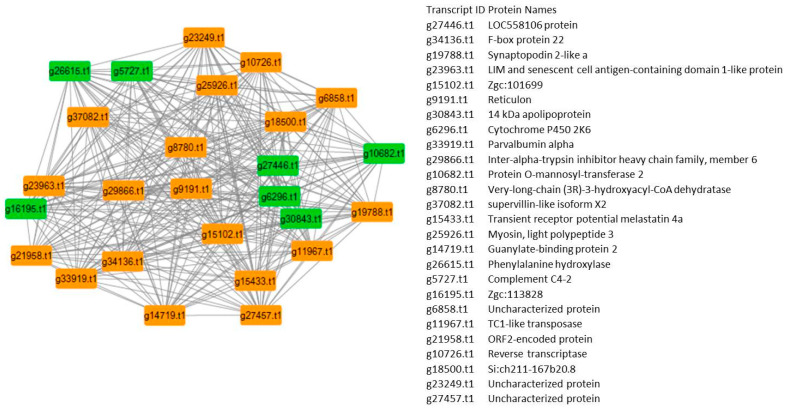
Protein–protein interaction network of control vs. treated. Different colors indicate different components identified in the gene lists.

**Figure 10 ijms-21-08180-f010:**
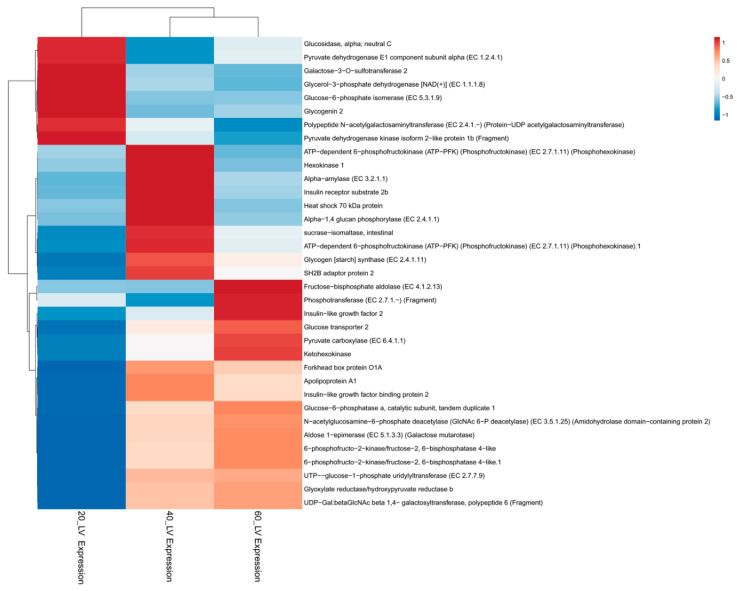
Heat map for hierarchical clustering of differential gene expression for selected genes. Fish feed 20% CHO was used as the control, and we also used 40% CHO and 60% CHO. Different colors indicate different *P*-value (−1 to 1).

**Table 1 ijms-21-08180-t001:** Statistics of *Labeo rohita* liver tissue transcriptome sequences.

Sample Name	Raw Reads	Processed Reads	% of Aligned Reads
60A_LV	35,558,985	30,800,241	89.58%
20_LV	23,396,281	21,202,572	92.89%
40_LV	24,004,149	21,608,231	89.84%

**Table 2 ijms-21-08180-t002:** Differential gene expression statistics for (**A**) 20_LV-Vs-40_LV and (**B**) 20_LV-Vs-60_LV.

**(A)**				
**Differential Gene Expression Statistics**				
**Summary**	**Total**	**Up**	**Down**	**Neutral**
No. of transcripts expressed in both samples	15,232	4464	4343	6425
No. of transcripts expressed only in 20_LV	3653	NA	NA	NA
No. of transcripts expressed only in 40_LV	2814	NA	NA	NA
No. of P significant transcripts	574	408	166	0
No. of Q significant transcripts	0	0	0	0
**(B)**				
**Differential Gene Expression Statistics**				
**Summary**	**Total**	**Up**	**Down**	**Neutral**
No. of transcripts expressed in both samples	15,360	4478	4171	6711
No. of transcripts expressed only in 20_LV	3525	NA	NA	NA
No. of transcripts expressed only in 60A_LV	2840	NA	NA	NA
No. of P significant transcripts	633	432	201	0
No. of Q significant transcripts	0	0	0	0

**Table 3 ijms-21-08180-t003:** The hub genes and their roles, and functional evidence.

Sl. No.	Hub genes	Function	Reference
1	*Synaptopodin*	Down-regulation of Synaptopodin is known to facilitate glycogen accumulation.	[26]
2	*LIM and senescent cell antigen-containing domain 1-like protein*	LIM and senescent cell antigen-containing domain 1-like protein (fragment) is down-regulated to increase abiogenesis.	[27]
3	*Reticulon*	Regulates fragmentation of ER during starvation but when glucose is in excess.	[28]
4	*Apolipoprotein*	ApoA-IV improves glucose homeostasis by promoting insulin secretion at high levels of glucose.	[29]
5	*Cytochrome P450*	Improves glucose homeostasis.	[30]
6	*Parvalbumin alpha*	Regulates brain–liver circuit for glucose homeostasis.	[31]
7	*Protein O-mannosyl-transferase 2*	Involved in biosynthesis of glycopeptides.	[32]
8	*Supervillin-like isoform X2*	Controls myogenesis and contributes to myogenic membrane structure and differentiation.	[33]
9	*Transient receptor potential melastatin 4a*	Plays role as gatekeeper in transepithelial Mg^2+^ transport to maintain Mg^2+^ homeostasis.	[34]
10	*Myosin, light polypeptide 3, skeletal muscle*	Controls oxidative glycolytic pathway in carbohydrate metabolism.	[35]
11	*Guanylate-binding protein 2*	Plays role in increasing activation of AMPK-p53 pathway and β-galactosidase.	[36]
12	*Phenylalanine hydroxylase*	Controls insulin secretion and glucose transport.	[37]
13	*Complement C4-2*	Associated with glucose intolerance.	[38]
14	*TC1-like transposase*	Reported to be associated with down-regulation of glycogenesis.	[39]
15	*Coagulation factor IXb (Zgc:136807)*	Reported to be associated with carbohydrate deficiency.	[40]
16	*Complement factor B/C2A*	Elevated level in adipose tissue leads to redistribution of visceral to subcutaneous fat and insulin resistance.	[41]
17	*Phospholipid-transporting ATPase (EC 7.6.2.1)*	Controls efficiency of transport of dietary fatty acids and lipids in fish.	[42]
18	*Alpha-2-macroglobulin (Fragment)*	Primarily associated with immunity by remodeling of liver metabolism in fish. It controls activation of the PI3K/Akt, ERK1, and MAPK pathways leading to mobilization of energetic resources away from growth and protein synthesis.	[43]
19	*S-adenosylmethionine synthase (EC 2.5.1.6)*	Associated with BMI and adiposity.	[44]
20	*Thrombospondin 2b*	Associated with fatty acid uptake by adipocytes.	[45]
21	*Notchless homolog 1 (Drosophila)*	Notch signaling central hub gene controlling glucose and lipid metabolism by insulin; also involved in inhibition of liver glucose production, including glycogenolysis and gluconeogenesis.	[46]
22	*Very-long-chain (3R)-3-hydroxyacyl-CoA dehydratase (EC 4.2.1.134)*	Involved in lipid biosynthesis especially in elongation of very-long-chain fatty acids.	[47]
23	*Obscurin-like 1b*	In fish myofibrillar component genes, obscurin is up-regulated with growth.	[48]
24	*Sk-tropomodulin (tropomodulin 4 (muscle))*	Regulates thin filament lengths in muscles.	[49]

**Table 4 ijms-21-08180-t004:** Depicts the ingredients of different levels of carbohydrate treatments.

Ingredients	Treatment_1 (20%)	Treatment_2 40%	Treatment_3 60%
Fish Meal	20	20	20
Casein	10.2	10.2	10.2
Gelatinized Starch	20	40	60
Glucose	0	0	0
Cellulose	40	20	0
Fish Oil	4	4	4
Vegetable Oil	3	3	3
Carboxy methyl cellulose (Binder)	0.5	0.5	0.5
Mineral Mix	1	1	1
Vitamin Mix	1	1	1
Vitamin C	0.2	0.2	0.2
Butylhydroxytoluene (BHT)	0.1	0.1	0.1
	100	100	100

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
