# Peer review of "Revealing Alteration in the Hepatic Glucose Metabolism of Genetically Improved Carp, Jayanti Rohu Labeo rohita Fed a High Carbohydrate Diet Using Transcriptome Sequencing"

_ijms, 2020, doi:10.3390/ijms21218180_

Round 1

Reviewer 1 Report

All contingencies have been addressed in the revised manuscript.

Recommendation: Acceptance

Author Response

Thank you for your recommendation of manuscript for acceptance. 

Reviewer 2 Report

This manuscript present the results of a thoroughly conducted research work on the dietary modulation of  fish liver Carbohydrate metabolism and gene expression for key metabolic enzymes.

The manuscript is well written, but some improvements are required to improve the clarity of the presentation and the text.

The 2nd paragraph of the introduction should be rephrased, start from the general (e.g animal/human level) and gradually focus on the specifics of  carbohydrate metabolism fish in general and specifically in the liver.

A reference to the effect of seasonal changes in temperature on the metabolic pathways, lipid and CHO metabolism should be included in this section.

The last part of the introduction should clearly support the need to cary the present work (which should be easy to do). Avoid the word “attempted” and bold statements such as “tolerance level”

The presentation of the work would benefit by some improvements, please 1st clarify what each investigated parameter was aiming to offer and support this with your  figures. For example I failed to see what is the difference between blue and Navy blue colour in Figure 4.

In the same manner, please include description of the content of all figures. For example see figure legend of Figure 7.

Page 14. The reference 83 does not support the otherwise correct 1st paragraph. Ibarz et al., did not investigate the effect of different dietary composition.

A few lines with the limitations of the present work should be included, for example possible changes in enzyme isoforms or effect of temperature or oxygen levels on the metabolism of fish, or an indirect effect of other parameters which contributed in the overall response (could be that a particular genetic profile make fish less prone to stress (e.g domestication)  and in turn this may affect the activation of specific metabolic pathways).

Author Response

To

The Reviewer

We have revised/corrected manuscript as per suggestions and based on valuable comments offered by the you. We are submitting herewith a list of responses to the each comments offered by the you.

I would be grateful if our revised manuscript could be considered for publication. 

Thanking you and with kind regards,

Yours sincerely,

Jitendra Kumar Sundaray,Ph.D

Principal Scientist and Head, Fish Genetics and Biotechnology Division,

ICAR - Central Institute of Freshwater Aquaculture,Bhubaneswar 751 002,

Odisha, India

Tel: +916742465407; +916742465414

Fax: +916742465407

Response to comments 

This manuscript present the results of a thoroughly conducted research work on the dietary modulation of fish liver Carbohydrate metabolism and gene expression for key metabolic enzymes.

 The manuscript is well written, but some improvements are required to improve the clarity of the presentation and the text.

 The 2nd paragraph of the introduction should be rephrased, start from the general (e.g animal/human level) and gradually focus on the specifics of carbohydrate metabolism fish in general and specifically in the liver.

 Compliance: As per suggestion we have modified the 2nd paragraph and included (Line 38-60)

A reference to the effect of seasonal changes in temperature on the metabolic pathways, lipid and CHO metabolism should be included in this section.

  Compliance: We have included this suggested part with suitable references. (Line 60-61)

The last part of the introduction should clearly support the need to carry the present work (which should be easy to do). Avoid the word “attempted” and bold statements such as “tolerance level”

Compliance: We have modified the content and deleted suggested words. (Line 77-86)

 The presentation of the work would benefit by some improvements, please 1st clarify what each investigated parameter was aiming to offer and support this with your figures. For example I failed to see what is the difference between blue and Navy blue colour in Figure 4.

Compliance: we have given details of each figure with legends

In the same manner, please include description of the content of all figures. For example see figure legend of Figure 7.

 Compliance:we have given details of each figure with legends

Page 14. The reference 83 does not support the otherwise correct 1st paragraph. Ibarz et al., did not investigate the effect of different dietary composition.

Compliance: We have removed this reference. 

A few lines with the limitations of the present work should be included, for example possible changes in enzyme isoforms or effect of temperature or oxygen levels on the metabolism of fish, or an indirect effect of other parameters which contributed in the overall response (could be that a particular genetic profile make fish less prone to stress (e.g domestication)  and in turn this may affect the activation of specific metabolic pathways).

Compliance:As suggested, we have included sentence of impact of temperature and oxyegen level on fish metabolism with suitable references and also discussed in the discussion and conclusion section, which is lacking in this work.

Round 2

Reviewer 2 Report

The revised version is improved, nevertheless the new references should be cited according to the instructions and the last paragraph of the Introduction needs revision. I suugest to remove the last sentence and revise the rest, 1)Include a citation, 2)support the present research work and the conclusion

Some of the new lines in the introduction,  need  reference to support them (e.g the assumption on the capacity for Cho metabolism of this fish) 

I wonder if the last sentence of the conclusion could serve as a base for the title and if possible revise both the title and the concluding paragraph. 

Author Response

The revised version is improved, nevertheless the new references should be cited according to the instructions and the last paragraph of the Introduction needs revision. I suugest to remove the last sentence and revise the rest, 1) Include a citation, 2)support the present research work and the conclusion

Compliance:  We have removed last sentence and included citation as per journal style.

Some of the new lines in the introduction,  need  reference to support them (e.g the assumption on the capacity for Cho metabolism of this fish) 

Compliance:  We have cited 2 review paper to support our assumption.

Polakof S, Panserat S, Soengas JL, Moon TW. Glucose metabolism in fish: a review. J Comp Physiol B. 2012 Dec;182(8):1015-45. doi: 10.1007/s00360-012-0658-7. Epub 2012 Apr 5. PMID: 22476584.

Hemre et al Carbohydrates in fish nutrition: effects on growth, glucose metabolism and hepatic enzymes, Aquaculture Nutrition 2002 8;175^194 10.1046/j.1365-2095.2002.00200.x

I wonder if the last sentence of the conclusion could serve as a base for the title and if possible revise both the title and the concluding paragraph. 

Compliance:  We have modified the title of manuscript and concluding paragraph as per suggestion:

Revealing alteration in the hepatic glucose metabolism of genetically improved carp, Jayanti rohu Labeo rohita fed a high carbohydrate diet using transcriptome sequencing